# How Artificial Intelligence Is Shaping Medical Imaging Technology: A Survey of Innovations and Applications

**DOI:** 10.3390/bioengineering10121435

**Published:** 2023-12-18

**Authors:** Luís Pinto-Coelho

**Affiliations:** 1ISEP—School of Engineering, Polytechnic Institute of Porto, 4200-465 Porto, Portugal; lfc@isep.ipp.pt; 2INESCTEC, Campus of the Engineering Faculty of the University of Porto, 4200-465 Porto, Portugal

**Keywords:** artificial intelligence, medical imaging, review, diagnostics, segmentation, classification

## Abstract

The integration of artificial intelligence (AI) into medical imaging has guided in an era of transformation in healthcare. This literature review explores the latest innovations and applications of AI in the field, highlighting its profound impact on medical diagnosis and patient care. The innovation segment explores cutting-edge developments in AI, such as deep learning algorithms, convolutional neural networks, and generative adversarial networks, which have significantly improved the accuracy and efficiency of medical image analysis. These innovations have enabled rapid and accurate detection of abnormalities, from identifying tumors during radiological examinations to detecting early signs of eye disease in retinal images. The article also highlights various applications of AI in medical imaging, including radiology, pathology, cardiology, and more. AI-based diagnostic tools not only speed up the interpretation of complex images but also improve early detection of disease, ultimately delivering better outcomes for patients. Additionally, AI-based image processing facilitates personalized treatment plans, thereby optimizing healthcare delivery. This literature review highlights the paradigm shift that AI has brought to medical imaging, highlighting its role in revolutionizing diagnosis and patient care. By combining cutting-edge AI techniques and their practical applications, it is clear that AI will continue shaping the future of healthcare in profound and positive ways.

## 1. Introduction

Advancements in medical imaging and artificial intelligence (AI) have ushered in a new era of possibilities in the field of healthcare. The fusion of these two domains has revolutionized various aspects of medical practice, ranging from early disease detection and accurate diagnosis to personalized treatment planning and improved patient outcomes.

Medical imaging techniques such as computed tomography (CT), magnetic resonance imaging (MRI), and positron emission tomography (PET) play a pivotal role in providing clinicians with detailed and comprehensive visual information about the human body. These imaging modalities generate vast amounts of data that require efficient analysis and interpretation, and this is where AI steps in.

AI, particularly deep learning algorithms, has demonstrated remarkable capabilities in extracting valuable insights from medical images [1]. Deep learning models, trained on large datasets, are capable of recognizing complex patterns and features that may not be readily discernible to the human eye [2,3]. These algorithms can even provide a new perspective about what image features should be valued to support decisions [4]. One of the key advantages of AI in medical imaging is its ability to enhance the accuracy and efficiency of disease diagnosis [1,5]. Through this process, AI can assist healthcare professionals in detecting abnormalities, identifying specific structures, and predicting disease outcomes [5,6].

By leveraging machine learning algorithms, AI systems can analyze medical images with speed and precision, aiding in the identification of early-stage diseases that may be difficult to detect through traditional methods. This early detection is crucial as it can lead to timely interventions, potentially saving lives and improving treatment outcomes [1,2,3].

Furthermore, AI has opened up new possibilities in image segmentation and quantification. By employing sophisticated algorithms, AI can accurately delineate structures of interest within medical images, such as tumors, blood vessels, or cells [7,8,9]. This segmentation capability is invaluable in treatment planning, as it enables clinicians to precisely target areas for intervention, optimize surgical procedures, and deliver targeted therapies [10].

The integration of AI and medical imaging has also facilitated the development of personalized medicine. Through the analysis of medical images and patient data, AI algorithms can generate patient-specific insights, enabling tailored treatment plans that consider individual variations in anatomy, physiology, and disease characteristics. This personalized approach to healthcare enhances treatment efficacy and minimizes the risk of adverse effects, leading to improved patient outcomes and quality of life [1,11,12].

Additionally, AI has paved the way for advancements in image-guided interventions and surgical procedures. By combining preoperative imaging data with real-time imaging during surgery, AI algorithms can provide surgeons with augmented visualization, navigation assistance, and decision support. These tools enhance surgical precision, reduce procedural risks, and enable minimally invasive techniques, ultimately improving patient safety and surgical outcomes [13].

Recently several cutting-edge articles have been published covering a wide variety of topics within the scope of medical imaging and AI. Many of these outstanding advancements are directed to cancer, a major cause of severe disease and mortality. The main contributions and fields will be addressed in the next sections.

## 2. Methodology

The primary aim of this review is to present a comprehensive overview of the influential artificial intelligence (AI) technological advancements that are shaping the landscape of medical imaging in recent years. The construction of the article dataset followed a two-stage methodology. Initially, to identify the most pertinent AI-supported clinical imaging application, searches were conducted on major scientific article repositories. In July 2023, queries were made on PubMed, IEEE, Scopus, ScienceDirect, Web of Science, and ACM, focusing on the Title and Abstract of articles. Filters for language (English only) and year of publication (2017 and onwards) were applied. Search terms encompassed key machine learning words and expressions (e.g., “machine learning”, “artificial intelligence”, “classification”, “segmentation”) combined with clinical image-related keywords (e.g., “image”, “pixel”, “resolution”, “MRI”, “PET”, “CT”). After article retrieval, duplicates were eliminated. It is also important to mention that preprint articles, such as arXiv, bioRxiv, medRxiv, among others, were also queried as part of the Scopus indexing system. These are major open-access article archives holding highly relevant manuscripts (considering the number of citations and widespread usage) but whose content was not peer reviewed.

In the second stage, the previously identified papers and their references were utilized as seeds to construct connection maps, employing the LitMaps [14] web tool to identify the most relevant technologies. The Iramuteq software [15] was also used to generate and explore word and concept networks using some of the included natural language processing tools [16]. The selection of technologies was based on manual observation of connection maps, with a focus on identifying healthcare-related keyword groups. The use of this methodology implied some ad hoc criteria since the mentioned tools are agnostic to the underlying clinical processes and not always are able to correctly group medical areas. With the described methodology, the ultimate aim was to encompass a broad spectrum of disease handling processes and support activities, emphasizing the most promising technological approaches to date while acknowledging identified limitations. Additionally, emphasis has been given to review articles that were specifically referenced when available for specific domains, as they offer an enhanced overview within a confined area of knowledge. The final article corpus showed a distribution by year of publication as depicted in Figure 1. It can be observed that 2023 has the highest number of review/survey articles, which can evidence the interest in the area but can also be an indicator of the diversity of involved technologies, demanding for an overview article.

## 3. Technological Innovations

Mathematical models and algorithms stand at the forefront of scientific exploration, serving as powerful tools that enable us to unravel complex phenomena, make predictions, and uncover hidden patterns in vast datasets. These essential components of modern research have not only revolutionized our understanding of the natural world but have also played a pivotal role in driving technological breakthroughs that open up numerous application possibilities across various domains. The synergy between mathematical models and algorithms has not only enhanced our understanding of the world but has also been a driving force behind technological advancements that have transformed our daily lives.

The earliest multilayer perceptron networks, while representing a crucial step in the evolution of neural networks, had notable limitations. One of the primary constraints was their shallow architecture, which consisted of only a few layers, limiting their ability to model complex patterns. Besides the model expansion restrictions imposed by the limited computing power, training these networks with multiple layers was also challenging. In particular, the earliest activation functions used in neural networks, including the sigmoid and hyperbolic tangent (tanh), led to the vanishing gradient problem [17] as their gradients became exceedingly small as inputs moved away from zero. This issue impeded the efficient propagation of gradients during training, resulting in slow convergence or training failures. Furthermore, the limited output range of these functions and their symmetric nature constrained the network’s ability to represent complex, high-dimensional data. Additionally, the computational complexity of these functions, particularly the exponential calculations, hindered training and inference in large networks. These shortcomings led to the development and widespread adoption of more suitable activation functions, such as the rectified linear unit (ReLU) [18] and its variants, which successfully addressed these issues and became integral components of modern deep learning architectures [19]. For these reasons, early multilayer perceptron networks struggled to capture complex patterns in data, making them unsuitable for tasks requiring the modeling of intricate relationships, ultimately leading to the necessity of exploration of more advanced architectures and training techniques.

Improvements in the artificial neurons’ functionality, more advanced architectures, and improved training algorithms supported by graphical computational units (GPU) came to open promising possibilities. The LeNet-5 architecture, developed for the recognition of handwritten digits [20], is a fundamental milestone for convolutional neural networks (CNNs) [21,22].

CNNs, inspired by the biological operation of animals’ vision system, assume that the input is the representation of image data. Current architectures follow a structured sequence of layers, each with specific functions to process and extract features from the input data [23]. The journey begins with the input layer, which receives raw image data, typically represented as a grid of pixel values, often with three color channels (red, green, blue) for color images. Following the input layer, the network employs convolutional layers, which are responsible for feature extraction. These layers use convolutional operations (of several types [22]) to detect local patterns and features in the input data. Early convolutional layers focus on detecting basic features like edges, corners, and textures. After each convolution operation, activation layers with rectified linear unit (ReLU) activation functions are applied to introduce nonlinearity. ReLU units help the network learn more complex patterns and enhance its ability to model the data effectively. Pooling (Subsampling) layers come next, reducing the spatial dimensions of the feature maps while preserving important information. Max pooling and average pooling are common operations that help make the network more robust to variations in scale and position. The sequence of convolutional layers continues, with additional layers stacked to capture increasingly complex and abstract features. These deeper layers are adept at detecting higher-level patterns, shapes, and objects in the data. Similar to the earlier convolutional layers, activation layers with ReLU functions are applied after each convolution operation, maintaining nonlinearity and enhancing feature learning. Pooling (subsampling) layers may be used again, further decreasing the spatial dimensions of the feature maps and retaining essential information. At the end of this sequence, after the network has extracted the most relevant information from the input data, a special set of vectors are obtained, designated by deep features [24]. These, located deep in the network, distill data into compact, meaningful forms that are highly discriminative. Or, in other words, after the progressive extraction of information, layer after layer, raw input data is refined into more condensed and abstract representations that are imbued with semantic meaning, encapsulating essential characteristics of the input. They are highly discriminative and have lower dimensionality than the raw input data, which not only conserves computational resources but also simplifies subsequent processing, making it especially beneficial in the analysis of high-dimensional data, such as images. This process also eliminates the tedious and error-prone process of handcrafted feature selection, leading to optimized feature sets and to the possibility of building the so-called “end-to-end” systems. Deep features can also help mitigate overfitting, a common challenge in machine learning, since by learning relevant representations, they prevent models from memorizing the training data and encourage more robust generalization.

Another great advantage of deep feature extraction pipelines is the possibility of using transfer learning techniques. In this case, a deep feature extraction network previously successfully developed on one task or dataset can be transferred and fine-tuned to another related task, significantly reducing the need for large, labeled datasets and speeding up model training. This versatility is a game changer in many applications.

After this extraction front end, continuing with the processing pipeline and moving towards the end of the network, fully connected layers are introduced. These layers come after the convolutional and pooling layers and play a pivotal role in feature aggregation and classification. The deep features extracted by the previous layers are flattened and processed through one or more fully connected layers.

Finally, the output layer emerges as the last layer of the network. The number of neurons in this layer corresponds to the number of classes in a classification task or the number of output units in a regression task. For classification tasks, a sigmoid or a softmax activation function is typically used to calculate class probabilities, providing the final output of the CNN [25,26]. A sigmoid function is commonly employed in binary classification, producing a single probability score indicating the likelihood of belonging to the positive class. The softmax function is favored for its ability to transform raw output scores into probability distributions across multiple classes. This conversion ensures that the computed probabilities represent the likelihood of the input belonging to each class, with the sum of probabilities equating to one, thereby constituting a valid probability distribution. Beyond this interpretability, both functions are differentiable, a critical attribute for the application of gradient-based optimization algorithms like backpropagation during training.

The described structured sequence of layers, from the input layer to the output layer, captures the hierarchical feature learning process in a CNN, allowing it to excel in image classification tasks (among others). Specific CNN architectures may introduce variations, additional components, or specialized layers based on the network’s design goals and requirements.

### 3.1. Transformers

CNNs are well suited for grid-like data, such as images, where local patterns can be captured efficiently. However, they struggle with sequential data because they lack a mechanism for modeling dependencies between distant elements (for example, in distinct time instants or far in the image). Also, CNNs do not inherently model the position or order of elements within the data. They rely on shared weight filters, which makes them translation invariant but can be problematic when absolute spatial relationships are important [27]. To overcome these limitations (handling sequential data, modeling long-range dependencies, incorporating positional information, and addressing tasks involving multimodal data, among others), transformers were introduced [28]. In the context of machine learning applied to images, transformers are a type of neural network architecture that extends the transformer model, originally designed for natural language processing [28], to handle computer vision tasks. These models are often referred to as vision transformers (ViTs) or image transformers [29] and come to introduce performance benefits, especially in noisy conditions [30,31]. In clinical settings, applications cover diagnosis and prognosis [32], encompassing classification, segmentation, and reconstruction tasks in distinct stages [31,33].

In vision transformers (ViT), the initial image undergoes a transformation process, wherein it is divided into a sequence of patches, as can be observed in Figure 2. Each of these patches is associated with a positional encoding technique, which captures and encodes the spatial positions of the patches, thus preserving spatial information. These patches, together with a class token, are then input into a transformer model to perform multi-head self-attention (MHSA) and generate embeddings that represent the learned characteristics of the patches. The class token’s state in the ViT’s output underscores a pivotal aspect of the model’s architecture since it acts as a global aggregator of information from all patches, offering a comprehensive representation of the entire image. The token’s state is dynamically updated during processing, reflecting a holistic understanding that encapsulates both local details and also the broader context of the image. Finally, a multilayer perceptron (MLP) is employed for the purpose of classifying the learned image representation. Notably, in addition to using raw images, it is also possible to supply feature maps generated by convolutional neural networks (CNNs) as input into a vision transformer for the purpose of establishing relational mappings [34]. It is also possible to use the transformer’s encoding technique to explore the model’s explainability [35].

The attention mechanism is a fundamental component in transformers. It plays a pivotal role in enabling the model to selectively focus on different parts of the input data with varying degrees of attention. At its core, the attention mechanism allows the model to assign varying levels of importance to different elements within the input data. This means the model can “pay attention” to specific elements while processing the data, prioritizing those that are most relevant to the task at hand. This selective attention enhances the model’s ability to capture essential information and relationships within the input. The mechanism operates as follows: First, the input data is organized into a sequence of elements, such as tokens in a sentence for NLP or patches in an image for computer vision. Then, the mechanism introduces three sets of learnable parameters: query (Q), key (K), and value (V). The query represents the element of interest, while the key and value pairs are associated with each element in the input sequence. For each element in the input sequence, the attention mechanism calculates an attention score, reflecting the similarity between the query and the key for that element. The method used to measure this similarity can vary, with techniques like dot product and scaled dot product being common choices. These attention scores represent how relevant each element is to the query. The next step involves applying the softmax function to the attention scores. This converts them into weights that sum to one, effectively determining the importance of each input element concerning the query. The higher the weight, the more attention the model allocates to that specific element in the input data. Finally, the attention mechanism computes a weighted sum of the values, using the attention weights. The resulting output is a combination of information from all input elements, with elements more relevant to the query receiving higher weight in the final representation [36,37].

The attention mechanism can be used in various ways (attention gate [38], mixed attention [39], among others in the medical field), with one prominent variant being self-attention. In self-attention, the query, key, and value all originate from the same input sequence. This allows the architecture to model relationships and dependencies between elements within the same sequence, making it particularly useful for tasks that involve capturing long-range dependencies and context [7,40,41].

The original ViT architecture, as in Figure 3a, was enhanced with the hierarchical vision transformer using shifted windows (SWIN transformer) [42] where a hierarchical partitioning of the image into patches is used. This means that the image is first divided into smaller patches, and then these patches are merged together as the network goes deeper, as in Figure 3b. This hierarchical approach allows SWIN to capture both local and global features in the image, which can improve its performance on a variety of tasks. In the SWIN transformer, images of different resolutions belonging to outputs of different stages can be used to facilitate segmentation tasks.

Another key difference between SWIN and ViT is that SWIN uses a shifted window self-attention mechanism, as depicted in Figure 4. This means that the self-attention operation is only applied to a local window of patches, or in other words, to a limited number of neighbor patches (as represented in green in Figure 4) rather than the entire image. Then, in a second stage, the attention window focus location is shifted to a different location (by patch cyclic shifting). This shifted window approach comes to reduce the computational load and complexity of the self-attention operation, which can improve the efficiency of the SWIN architecture. These differences, when compared with the original ViT, allow a more efficient and scalable architecture, which were further refined in SWIN v2 [43].

The transformer-based approach has received a lot of attention due to its effectiveness, still with improvement opportunities [44]. The described innovations have been crucial in advancing the state of the art in medical image processing, covering machine learning tasks, such as classification, segmentation, synthesis (image or video), detection, and captioning [34,45]. By enhancing the model’s ability to focus on relevant information and understand complex relationships within the data, the attention mechanism represents a significant step in the improvement of the quality and effectiveness of various deep learning applications in the medical field.

Within the broad category of computer vision and artificial intelligence, the YOLO algorithm [46], which stands for “you only look once”, has gained a lot of popularity due to its performance in real-time object detection tasks. In the medical imaging field, the term “YOLO” is sometimes used more broadly to refer to implementations or systems that use one of the versions of the YOLO algorithm. It approaches object detection as a regression problem, predicting bounding box coordinates and class probabilities directly from the input image in a single pass through its underlying neural network (composed of backbone, neck, and head sections). This single-pass processing, where the image is divided into a grid for simultaneous predictions, distinguishes YOLO from other approaches and contributes to its exceptional speed. Postprediction, nonmaximum suppression is applied to filter redundant and low-confidence predictions, ensuring that each object is detected only once. In the medical field, YOLO has been used for a variety of imaging tasks, including cytology automation [47], detecting lung nodules in CT scans [48], segmentation of structures [49], detecting breast cancer in mammograms [50], or to track needles in ultrasound sequences [51], among others. YOLO’s fast and accurate object detection capabilities make it an excellent choice for many medical imaging applications.

Finally, it is noteworthy to highlight the emergence of hybrid approaches that combine the aforementioned algorithms, as observed in instances like TransU-net [52] or ViT-YOLO [53]. These combinations aim to leverage the strengths of each individual algorithm, with the objective of achieving performance enhancements. It is important to acknowledge, however, that these approaches are still in an early stage of development and are not explored here.

### 3.2. Generative Models

Generative models are a class of machine learning models that can generate new data based on training data. Other generative models include generative adversarial networks (GANs), variational autoencoders (VAEs), and flow-based models. Each can produce high-quality images.

Generative adversarial networks, or GANs, are a class of machine learning models introduced in 2014 [54] that excel at generating data, often in the form of images, but applicable to other data types like text or audio as well. GANs consist of two neural networks: a generator and a discriminator. The generator creates synthetic data from random noise and aims to produce data that are indistinguishable from real data, while the discriminator tries to distinguish between real and fake data, as represented in Figure 5. Through an adversarial training process, these networks compete, with the generator continually improving its ability to create realistic data and the discriminator enhancing its capacity to identify real from fake data.

GANs have revolutionized the field of data generation, a highly valued resource due to the data avidity of modern machine learning systems, due to the lack of data in some areas and due to data protection and security constraints. These networks offer a highly effective way to create synthetic data that closely resemble real data. This is highly valuable, especially when dealing with limited datasets, as GANs can help augment training data for various machine learning tasks. For instance, in medical imaging, where obtaining large, diverse datasets can be challenging, GANs enable researchers to generate additional, realistic medical images for training diagnostic models, ultimately improving the accuracy of disease detection [55]. A recent study by Armanious et al. proposed a new framework called MedGAN [56] for medical image-to-image translation that operates on the image level in an end-to-end manner. MedGAN builds upon recent advances in the field of GANs by merging the adversarial framework with a new combination of nonadversarial losses. The framework utilizes a discriminator network as a trainable feature extractor which penalizes the discrepancy between the translated medical images and the desired modalities. Style-transfer losses are also utilized to match the textures and fine structures of the desired target images to the translated images. Additionally, a new generator architecture, titled CasNet, enhances the sharpness of the translated medical outputs through progressive refinement via encoder–decoder pairs. MedGAN was applied to three different tasks: PET–CT translation, correction of MR motion artefacts, and PET image denoising. Perceptual analysis by radiologists and quantitative evaluations illustrate that MedGAN outperforms other existing translation approaches.

Generative adversarial networks (GANs) have been a promising tool in the field of medical image analysis [57], particularly in image-to-image translation. Skandarani et al. [58] conducted an empirical study on GANs for medical image synthesis. The results revealed that GANs are far from being equal as some are ill-suited for medical imaging applications while others are much better off. The top-performing GANs are capable of generating realistic-looking medical images by FID standards that can fool trained experts in a visual Turing test and comply with some metrics [58]. The introduction of these models into clinical practice has been cautious [59], but the advantages and performance that have been successively achieved with their development have allowed GANs to become a successful technology.

Along with GANs, variational autoencoders (VAEs) are a popular technique for image generation. While both models are capable of generating images, they differ in their approach and training methodology. VAEs are a type of generative model that learns to encode the fundamental information of the input data into a latent space. The encoder network maps the input data to a latent space, which is then decoded by the decoder network to generate the output image. VAEs are trained using a probabilistic approach that maximizes the likelihood of the input data given the latent space. VAEs are better suited for applications that require probabilistic modeling, such as image reconstruction and denoising. This approach is capable of generating high-quality images but may suffer from blurry outputs [60,61,62].

Diffusion models constitute another class of generative models employed in image synthesis, functioning by iteratively transforming a base distribution into a target distribution through a series of diffusion steps [63]. These models leverage the concept of image diffusion, wherein the generation process unfolds progressively by adding noise to the image iteratively. Typically, the generation process commences with a simple distribution, such as a Gaussian, and refines it over multiple steps to approximate the desired complex distribution of real images. The iterative nature of diffusion models allows them to capture intricate structures and nuanced details present in medical images, where they can outperform GAN [64,65]. They can also be applied to video data [66,67].

Flow-based generative models represent a distinct approach in variational inference and natural image generation, recently gaining attention in the realm of computer vision [68]. The foundational concept, introduced in [69], centers around the utilization of a (normalizing) flow—a sequence of invertible mappings—to construct the transformation of a probability density, approximating a posterior distribution. The process commences with an initial variable, progressively mapping it to a variable characterized by a simple distribution (such as an isotropic Gaussian). This is achieved by iteratively applying the change of variable rule, akin to the inference mechanism in an encoder network. In the context of image generation, the initial variable is the real image governed by an unknown probability function. Through the employment of a well-designed inference network, the flow undergoes training to learn an accurate mapping. Importantly, the invertibility of the flow-based model facilitates the straightforward generation of synthetic images. This is accomplished by sampling from the simple distribution and navigating through the map in reverse. Comparative to alternative generative models and autoregressive models, flow-based methods offer a notable advantage by enabling tractable and accurate log-likelihood evaluation throughout the training process [70]. Simultaneously, they afford an efficient and exact sampling process from the simple prior distribution during testing. Image modality transfer [71] and 3D data augmentation [72] are promising areas in the medical field. 

GANs are highly popular for magnetic resonance applications due to their ability to generate additional datasets and also due to the existing datasets that can support the training of effective models [73]. Reconstruction and segmentation tasks are also an important field of application. Here, the adversarial training plays a crucial role in imposing robust constraints on both the shape and texture of the generator’s output [73]. In some cases, GANs can be preferred over VAE due easier optimal model optimization [74]. In many applications, a balance must be found between the ability to generate high-quality samples, achieve fast sampling (inference), and exhibit mode diversity [75].

Overall, generative approaches are vital in machine learning for medical images due to their capacity to generate realistic data, drive innovation in image generation and manipulation, facilitate image-to-image translation, and open up creative opportunities for content generation across various domains.

### 3.3. Deep Learning Techniques and Performance Optimization

Medical imaging techniques are based on different physical principles, each with their benefits and limitations. The ability to deal with such diverse modalities is also an important aspect to be addressed by AI. In [76], a set of “tricks” are presented to improve the performance of deep learning models for multimodal image classification tasks. The authors start by emphasizing the increasing importance of multimodal image classification, which involves utilizing information from multiple modalities, such as images, text, and other data sources. For this, they also address the challenges specific to multimodal datasets, including data preprocessing, feature extraction, data imbalance, heterogeneity of modalities, data fusion, and model optimization. As defined by the authors, a “bag of tricks” or techniques can enhance the effectiveness of these models in handling multimodal data. These tricks can focus on the data, covering feature alignment, modality-specific preprocessing, and class balancing techniques, and also on the processing, using architectural modifications, training strategies, and regularization techniques. For the evaluation of such systems, benchmarking approaches are also presented and explored. These are valuable insights for researchers and practitioners working in the field of multimodal image classification.

## 4. Applications

AI-based imaging techniques can be divided in eight distinct categories: acquisition, preprocessing, feature extraction, registration, classification, object localization, segmentation, and visualization. These can also be organized in the clinical process pipeline broadly encompassing prevention, diagnostics, planning, therapy, prognostic, and monitoring. It is also possible to focus on the human organ or physiological process under focus. Using this last perspective, groups have been created using the associated keywords of the selected papers, and their relative expression has been calculated, as in Figure 6. Notably, lungs emerge as the primary focus, likely attributed to the aftermath of the recent COVID-19 pandemic and the availability of novel, untapped datasets. The significance of the affected organ in human life should also be a pivotal factor driving researchers’ interest in each domain.

### 4.1. Medical Image Analysis for Disease Detection and Diagnosis

Medical image analysis for disease detection and diagnosis is a rapidly evolving field that holds immense potential for improving healthcare outcomes. By harnessing advanced computational techniques and machine learning algorithms, medical professionals are now able to extract invaluable insights from various medical imaging modalities [76,77].

Artificial intelligence is an area where great progress has been observed, and the number of techniques applicable to medical image processing has been increasing significantly. In this context of diversity, review articles where different techniques are presented and compared are useful. For example, in the area of automated retinal disease assessment (ARDA), AI can be used to help healthcare workers in the early detection, screening, diagnosis, and grading of retinal diseases such as diabetic retinopathy (DR), retinopathy of prematurity (RoP), and age-related macular degeneration (AMD), as shown in the comprehensive survey presented in [77]. The authors highlight the significance of medical image modalities, such as optical coherence tomography (OCT), fundus photography, and fluorescein angiography, in capturing detailed retinal images for diagnostic purposes and explain how AI can cope with these distinct information sources, either isolated or combined. The limitations and subjectivity of traditional manual examination and interpretation methods are emphasized, leading to the exploration of AI-based solutions. For this, an overview of the utilization of deep learning models is presented, and the most promising results in the detection and classification of retinal diseases, including age-related macular degeneration (AMD), diabetic retinopathy, and glaucoma, are thoroughly covered. The role of AI in facilitating the analysis of large-scale retinal datasets and the development of computer-aided diagnostic systems is also highlighted. However, AI is not always a perfect solution, and the challenges and limitations of AI-based approaches are also covered, addressing issues related to data availability, model interpretability, and regulatory considerations. Given the significant interest in this field and the promising results that AI has yielded, other studies have also emerged to cover various topics related to eye image analysis [78,79].

Another area of great interest is brain imaging, whose techniques play a crucial role in understanding the intricate workings of the human brain and in diagnosing neurological disorders. Methods such as magnetic resonance imaging (MRI), functional MRI (fMRI), positron emission tomography (PET), or electroencephalography signals (EEG) provide valuable insights into brain structure, function, and connectivity. However, the analysis of these complex data, be it images or signals, requires sophisticated tools and expertise. Again, artificial intelligence (AI) comes into play. The synergy between brain imaging and AI has the potential to revolutionize neuroscience and improve patient care by unlocking deeper insights into the intricacies of the human brain. In [80], a powerful combination of deep learning techniques and the sine–cosine fitness grey wolf optimization (SCFGWO) algorithm is used on the detection and classification of brain tumors. It addresses the importance of accurate tumor detection and classification as well as the associated challenges. Complexity and variability are tackled by convolutional neural networks (CNNs) that can automatically learn and extract relevant features for tumor analysis. In this case, the SCFGWO algorithm is used to fine-tune the parameters of the CNN leading to an optimized performance. Metrics, such as accuracy, sensitivity, specificity, and F1-score, are compared with other existing approaches to showcase the effectiveness and benefits of the proposed method in brain tumor detection and classification. The advantages and limitations of the proposed approach and the potential impact of the research in clinical practice are also mentioned.

Lung imaging has been a subject of extensive research interest [81,82], primarily due to the aggressive nature of lung cancer and its tendency to be detected at an advanced stage, leading to high mortality rates among cancer patients. In this context, accurate segmentation of lung fields in medical imaging plays a crucial role in the detection and analysis of lung diseases. In a recent study [83], the authors focused on segmenting lung fields in chest X-ray images using a combination of superpixel resizing and encoder–decoder segmentation networks. The study effectively addresses the challenges associated with lung field segmentation, including anatomical variations, image artifacts, and overlapping structures. It emphasizes the potential of deep learning techniques and the utilization of encoder–decoder architectures for semantic segmentation tasks. The proposed method, which combines superpixel resizing with an encoder–decoder segmentation network, demonstrates a high level of effectiveness compared to other approaches, as assessed using evaluation metrics such as the Dice similarity coefficient, Jaccard index, sensitivity, specificity, and accuracy.

More recently, the interest in lung imaging has been reinforced due to its importance in the diagnosis and monitoring of COVID-19 disease. In a notable study [84], the authors delve into the data-driven nature of AI and its need for high-quality data. They specifically focus on the generation of synthetic data, which involves creating artificial instances that closely mimic real data. In fact, by using the proposed approach, the synthetic images are nearly indistinguishable from read images when compared using the structural similarity index (SSIM), peak signal-to-noise ratio (PSNR), and the Fréchet inception distance (FID). In this case, lung CT for COVID-19 diagnosis is used as an application example where this proposed approach has shown to be successful. The problem is tackled by means of a new regularization strategy, which refers to a technique used to prevent overfitting in ML models. This strategy does not require making significant changes to the underlying neural network architecture, making it easier to implement. Furthermore, the proposed method’s efficacy extends beyond lung CT for COVID-19 diagnosis and can be easily adapted to other image types or imaging modalities. Consequently, future research endeavors can explore its applicability to diverse diseases and investigate its relevance to emerging AI topics, such as zero-shot or few-shot learning.

Breast cancer, the second most reported cancer worldwide, must be diagnosed as early as possible for a good prognostic. In this case, medical imaging is paramount for disease prevention and diagnosis. The effectiveness of an AI-based approach is evaluated in [85]. The authors present a novel investigation that constructs and evaluates two computer-aided detection (CAD) systems for digital mammograms. The objective was to differentiate between malignant and benign breast lesions by employing two state-of-the-art approaches based on radiomics (with features such as intensity, shape, and texture) and deep transfer learning concepts and technologies (with deep features). Two CAD systems were trained and assessed using a sizable and diverse dataset of 3000 images. The findings of this study indicate that deep transfer learning can effectively extract meaningful features from medical images, even with limited training data, offering more discriminatory information than traditional handcrafted radiomics features. However, explainability, a desired characteristic in artificial intelligence and in medical decision systems in particular, must be further explored to fully unravel the mysteries of these “black-box” models.

Still, concerning breast imaging, and addressing the typical high data needs of machine learning systems, a study was made to compare and optimize models using small datasets [86]. The article discusses the challenges associated with limited data, such as overfitting and model generalization. Distinct CNN architectures, such as AlexNet, VGGNet, and ResNet, are trained using small datasets. The authors discuss strategies to mitigate these limitations, such as data augmentation techniques, transfer learning, and model regularization. With these premises, a multiclass classifier, based on the BI-RADS lexicon on the INBreast dataset [87], was developed. Compared with the literature, the model was able to improve the state-of-the-art results. This comes to reinforce that discriminative fine-tuning works well with state-of-the-art CNN models and that it is possible to achieve excellent performance even on small datasets.

Radiomics and artificial intelligence (AI) play pivotal roles in advancing breast cancer imaging, offering a range of applications across the diagnostic spectrum. These technologies contribute significantly to risk stratification, aiding in the determination of cancer recurrence risks and providing valuable insights to guide treatment decisions [88,89]. Moreover, AI algorithms leverage radiomics features extracted from diverse medical imaging modalities, such as mammography, ultrasound, magnetic resonance imaging (MRI), and positron emission tomography (PET), to enhance the accuracy of detecting and classifying breast lesions [88,89]. For treatment planning, radiomics furnishes critical information regarding treatment effectiveness, facilitating the prediction of treatment responses and the formulation of personalized treatment plans [90]. Additionally, radiomics serves as a powerful tool for prognosis, enabling the prediction of outcomes such as disease-free survival and recurrence risk in breast cancer patients [91]. Furthermore, the robustness of MRI-based radiomics features against interobserver segmentation variability has been highlighted, indicating their potential for future breast MRI-based radiomics research [92].

Liver cancer is the third most common cause of death from cancer worldwide [93], and its incidence has been growing. Again, the development of the disease is often asymptomatic, making screening and early detection crucial for a good prognosis. In [8], the authors focus on the segmentation of liver lesions in CT images of the LiTS dataset [94]. As a novelty, the paper proposes an intelligent decision system for segmenting liver and hepatic tumors by integrating four efficient neural networks (ResNet152, ResNeXt101, DenseNet201, and InceptionV3). These classifiers are independently operated, and a final result is obtained by postprocess to eliminate artifacts. The obtained results were better than those obtained by the individual networks. In fact, concerning liver and pancreatic images, the use of AI algorithms is already a reality for speeding up repetitive tasks, such as segmentation, acquiring new quantitative parameters, such as lesion volume and tumor burden, improving image quality, reducing scanning time, and optimizing imaging acquisition [95].

Diabetic retinopathy (DR) is a significant cause of blindness globally, and early detection and intervention can help change the outcomes of the disease. AI techniques, including deep learning and convolutional neural networks (CNN), have been applied to the analysis of retinal images for DR screening and diagnosis [96]. Some studies have shown promising results in detecting referable diabetic retinopathy (rDR) using AI algorithms with high sensitivity and specificity compared to human graders [97], while reducing the associated human resources. For example, a study using a deep learning-based AI system achieved 97.05% sensitivity, 93.4% specificity, and 99.1% area under the curve (AUC) in classifying rDR as moderate or worse diabetic retinopathy, referable diabetic macular edema, or both [97]. Nevertheless, there are also shortcomings, such as the lack of standards for development and evaluation and the limited scope of application [98]. 

AI can also help in the detection and prediction of age-related macular degeneration (AMD). AI-based systems can screen for AMD and predict which patients are likely to progress to late-stage AMD within two years [99]. AI algorithms can provide analyses to assist physicians in diagnosing conditions based on specific features extrapolated from retinal images [100].

Yet in this area, optical coherence tomography (OCT) is a valuable tool in diagnosing various eye conditions and is where artificial intelligence (AI) can successfully be used. AI-assisted OCT has several advantages and applications in ophthalmology for diagnosis, monitoring, and disease-progression estimation (e.g., for glaucoma, macular edema, or age-related macular degeneration) [101]. AI-assisted OCT can provide more accurate and sensitive results compared to traditional methods [102]. For example, an OCT-AI-based telemedicine platform achieved a sensitivity of 96.6% and specificity of 98.8% for detecting urgent cases, and a sensitivity of 98.5% and specificity of 96.2% for detecting both urgent and routine cases [103].

These tools can lead to more efficient and objective ways of diagnosing and managing eye conditions.

### 4.2. Imaging and Modeling Techniques for Surgical Planning and Intervention

Imaging and 3D modeling techniques, coupled with the power of artificial intelligence (AI), have revolutionized the field of surgical planning and intervention, offering numerous advantages to both patients and healthcare professionals. By leveraging the capabilities of AI, medical imaging data, such as CT scans and MRI images, can be transformed into detailed three-dimensional models that provide an enhanced understanding of a patient’s anatomy. This newfound precision and depth of information allow surgeons to plan complex procedures with greater accuracy, improving patient outcomes and minimizing risks. Furthermore, AI-powered algorithms can analyze vast amounts of medical data, assisting surgeons in real-time during procedures, guiding them with valuable insights, and enabling personalized surgical interventions. For example, in [49], a new deep learning (DL)-based tool for segmenting anatomical structures of the left heart from echocardiographic images is proposed. It results from a combination of the YOLOv7 algorithm and U-net, specifically addressing segmentation of echocardiographic images into LVendo, LVepi, and LA.

Additionally, the integration of 3D printing technology with imaging and 3D modeling techniques further amplifies the advantages of surgical planning and intervention. With 3D printing, these intricate anatomical models can be translated into physical objects, allowing surgeons to hold and examine patient-specific replicas before the actual procedure. This tangible representation aids in comprehending complex anatomical structures, identifying potential challenges, and refining surgical strategies. Surgeons can also utilize 3D-printed surgical guides and implants, customized to fit each patient’s unique anatomy, thereby enhancing precision and reducing operative time.

These benefits are described and explored in [104], covering the operative workflow involved in the process of creating 3D-printed models of the heart using computed tomography (CT) scans. The authors begin by emphasizing the importance of accurate anatomical models in surgical planning, particularly in complex cardiac cases. They also discuss how 3D printing technology has gained prominence in the medical field, allowing for the creation of patient-specific anatomical models. In their developments, they thoroughly describe the operative workflow for generating 3D-printed heart models. Throughout the process, the challenges and limitations of the operative workflow from CT to 3D printing of the heart are covered. They also discuss factors such as cost, time, expertise required, and the need for validation studies to ensure the accuracy and reliability of the printed models.

A similar topic is presented in [105]. Here the authors focus specifically on coronary artery bypass graft (CABG) procedures and describe the feasibility of using a 3D modeling and printing process to create surgical guides, contributing to the success of the surgery and enhancing patient outcomes. In this paper, the authors also discuss the choice of materials for the 3D-printed guide, considering biocompatibility and sterility requirements. In addition, a case study that demonstrates the successful application of the workflow in a real clinical scenario is presented.

The combination of AI-driven imaging, 3D modeling, and 3D printing technologies revolutionizes surgical planning and intervention, empowering healthcare professionals with unparalleled tools to improve patient outcomes, create personalized solutions, and redefine the future of surgical practice. These advancements in imaging and 3D modeling techniques, driven by AI, are driving a new era of surgical precision and innovation in healthcare.

### 4.3. Image and Model Enhancement for Improved Analysis

Decision-making and diagnosis are important purposes for clinical applications, but AI can also play an important role in other applications of the clinical process. For example, in [106] the authors focus on the application of colorization techniques to medical images, with the goal of enhancing the visual interpretation and analysis by adding chromatic information. The authors highlight the importance of color in medical imaging as it can provide additional information for diagnosis, treatment planning, and educational purposes. They also address the challenges associated with medical image colorization, including the large variability in image characteristics and the need for robust and accurate colorization methods. The proposed method utilizes a spatial mask-guided colorization with a generative adversarial network (SMCGAN) technique to focus on relevant regions of the medical image while preserving important structural information during the process. The evaluation was based on a dataset from the Visible Human Project [107] and from the prostate dataset NCI-ISBI 2013 [108]. With the presented experimental setup and evaluation metrics used for performance assessment, the proposed technique was able to outperform the state-of-the-art GAN-based image colorization approaches with an average improvement of 8.48% in the peak signal-to-noise ratio (PSNR) metric.

In complex healthcare scenarios, it is crucial for clinicians and practitioners to understand the reasoning behind AI models’ predictions and recommendations. Explainable AI (XAI) plays a pivotal role in the domain of medical imaging techniques for decision support, where transparency and interpretability are paramount. In [9], the authors address the problem of nuclei detection in histopathology images, which is a crucial task in digital pathology for diagnosing and studying diseases. They specifically propose a technique called NDG-CAM (nuclei detection in histopathology images with semantic segmentation networks and Grad-CAM). Grad-CAM (gradient-weighted class activation mapping) [109] is a technique used in computer vision and deep learning to visualize and interpret the regions of an image that are most influential in the prediction made by a convolutional neural network. Hence, in the proposed methodology, the semantic segmentation network aims to accurately segment the nuclei regions in histopathology images, while Grad-CAM helps visualize the important regions that contribute to the model’s predictions, helping to improve the accuracy and interpretability of nuclei detection. The authors compare the performance of their method with other existing nuclei detection methods, demonstrating that NDG-CAM achieves improved accuracy while providing interpretable results.

Still with the purpose of making AI provide human understandable results, the authors in [110] focus on the development of an open-source COVID-19 CT dataset that includes automatic lung tissue classification for radiomics analysis. The challenges associated with COVID-19 research, including the importance of large-scale datasets and efficient analysis methods are covered. The potential of radiomics, which involves extracting quantitative features from medical images, in aiding COVID-19 diagnosis, prognosis, and treatment planning, are also mentioned. The proposed dataset consists of CT scans from COVID-19 patients, which are annotated with labels indicating different lung tissue regions, such as ground-glass opacities, consolidations, and normal lung tissue.

Novel machine learning techniques are also being used to enhance the resolution and quality of medical images [111]. These techniques aim to recover fine details and structures that are lost or blurred in low-resolution images, which can improve the diagnosis and treatment of various diseases. One of the novel machine learning techniques is based on GANs. For example, Bing at al. [112] propose the use of an improved squeeze-and-excitation block that selectively amplifies the important features and suppresses the nonimportant ones in the feature maps. A simplified EDSR (enhanced deep super-resolution) model to generate high-resolution images from low-resolution inputs is also proposed, along with a new fusion loss function. The proposed method was evaluated on public medical image datasets and compared with state-of-the-art deep learning-based methods, such as SRGAN, EDSR, VDSR, and D-DBPN. The results show that the proposed method achieves better visual quality and preserves more details, especially for high upscaling factors.

Vision transformers, with their ability to treat images as sequences of tokens and to learn global dependencies among them, can capture long-range and complex patterns in images, which can benefit super-resolution tasks. Zhu et al. [113] propose the use of vision transformers with residual dense connections and local feature fusion. This method proposes an efficient vision transformer architecture that can achieve high-quality single-image super-resolution for various medical modalities, such as MRI, CT, and X-ray. The key idea is to use residual dense blocks to enhance the feature extraction and representation capabilities of the vision transformer and to use local feature fusion to combine the low-level and high-level features for better reconstruction. Moreover, this method also introduces a novel perceptual loss function that incorporates prior knowledge of medical image segmentation to improve the image quality of desired aspects, such as edges, textures, and organs. In another work, Wei et al. [114] propose to adapt the SWIN transformer, which is a hierarchical vision transformer that uses shifted windows to capture local and global information, to the task of automatic medical image segmentation. The high-resolution SWIN transformer uses a U-net-like architecture that consists of an encoder and a decoder. The encoder converts the high-resolution input image into low-resolution feature maps using a sequence of SWIN transformer blocks, and the decoder gradually generates high-resolution representations from low-resolution feature maps using upsampling and skip connections. The high-resolution SWIN transformer can achieve state-of-the-art results on several medical image segmentation datasets, such as BraTS, LiTS, and KiTS (details below).

In addition, perceptual loss functions can be used to further enhance generative techniques. These are designed to measure the similarity between images in terms of their semantic content and visual quality rather than their pixel-wise differences. Perceptual loss functions can be derived from pretrained models, such as image classifiers or segmenters, that capture high-level features of images. By optimizing the perceptual loss functions, the super-resolution models can generate images that preserve the important structures and details of the original images while avoiding artifacts and distortions [112,115].

Medical images often suffer from noise, artifacts, and limited resolution due to the physical constraints of the imaging devices. Therefore, developing effective and efficient methods for medical image super-resolution is a challenging and promising research topic, searching to obtain previously unachievable details and resolution [116,117].

### 4.4. Medical Imaging Datasets

Numerous advancements outlined above have arisen through machine learning public challenges. These initiatives provided supporting materials in the form of datasets (which are often expensive and time consuming to collect) and, at times, baseline algorithms, contributing to the facilitation of various research studies aimed at the development and evaluation of novel algorithms. The promotion of a competitive objective was pivotal for promoting the development of a scientific community around a given topic. In Table 1, some popular datasets are presented.

## 5. Conclusions

Cutting-edge techniques that push the limits of current knowledge have been covered in this editorial. For those focused on the AI aspects of technology, evolutions have been reported in all stages of the medical imaging machine learning pipeline. As mentioned, the data-driven nature of these techniques requires that special attention is given to it. Beyond a high-quality dataset [110], attention can be given to the generation of more data [84] and better data [83]. The training process can be optimized to deal with small datasets [86], or techniques can be used to improve the parameter optimization process [80]. To better understand the models’ operating, we can use explainable AI techniques [9]. We can also focus on generating a better output by combining several classifiers [8] or by adding useful information, such as colors [106]. Many of the involved challenges throughout the process can address using a “bag of tricks” [76]. The advantages of using AI in medical imaging applications is explored in [77], and its ability to perform better than feature-based approaches is covered in [85]. Finally, applications of AI to 3D modeling and physical object generation are covered in [104,105].

The field of medical imaging and AI is evolving rapidly, driven by ongoing research and technological advancements. Researchers are continuously exploring novel algorithms, architectures, and methodologies to further enhance the capabilities of AI in medical imaging. Additionally, collaborations between clinicians, computer scientists, and industry professionals are vital in translating research findings into practical applications that can benefit patients worldwide.

In conclusion, the fusion of medical imaging and AI has brought about significant advancements in healthcare. From early disease detection to personalized diagnosis and therapy, AI has demonstrated its potential to revolutionize medical practice. By harnessing the power of AI, medical professionals can leverage the wealth of information contained within medical images to provide accurate diagnoses, tailor treatment plans, and improve patient outcomes. As technology continues to advance, we can expect even more groundbreaking innovations that will further transform the landscape of medical imaging and AI in the years to come.

## Figures and Tables

**Figure 1 bioengineering-10-01435-f001:**
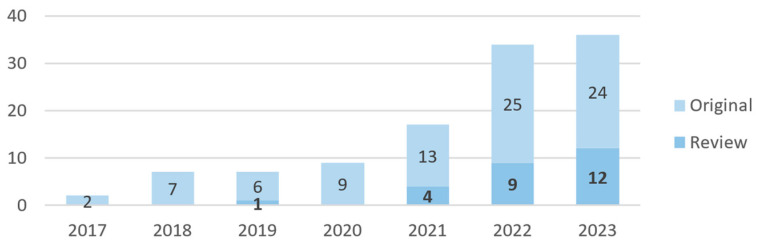
Distribution of the selected articles by year of publication.

**Figure 2 bioengineering-10-01435-f002:**
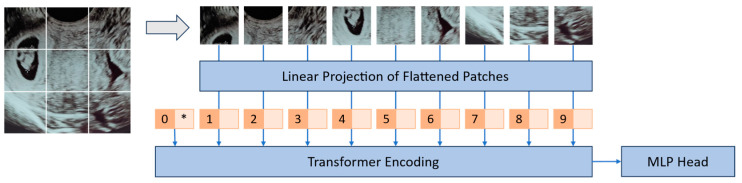
Pipeline for applying the transformer’s technique to images.

**Figure 3 bioengineering-10-01435-f003:**
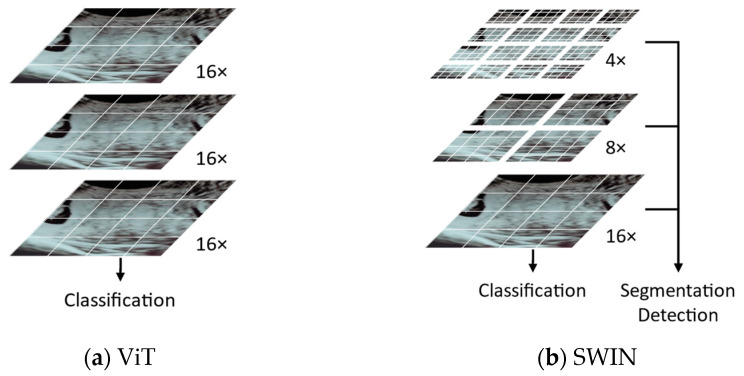
Comparison of architecture operation when going deep in the network.

**Figure 4 bioengineering-10-01435-f004:**
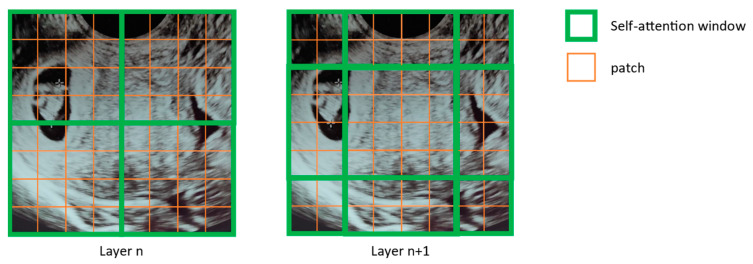
Shifted window’s mechanism for the self-attention mechanism in the SWIN transformer.

**Figure 5 bioengineering-10-01435-f005:**
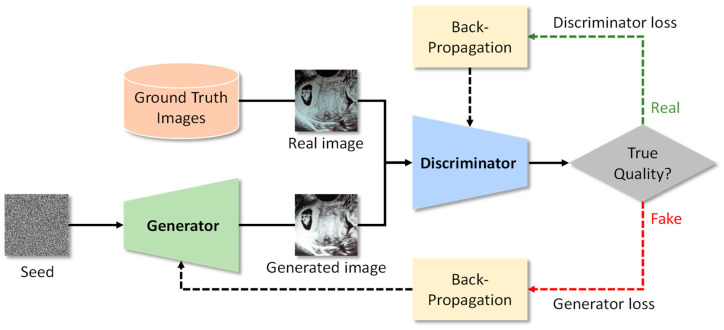
Architecture overview for a generative adversarial network for images.

**Figure 6 bioengineering-10-01435-f006:**
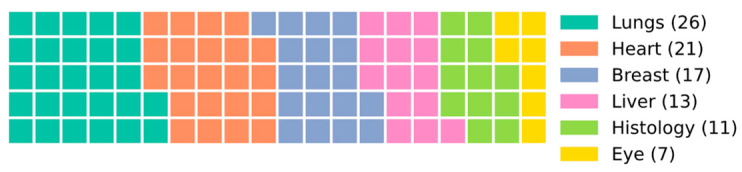
Number of publications per area of interest (showing the first six highest ranked, normalized to 100).

**Table 1 bioengineering-10-01435-t001:** Examples of datasets with medical images.

Name	Description	Reference
BRATS	The Multimodal Brain Tumor Segmentation Benchmark (BRATS) is an annual challenge that aims to compare different algorithms for brain tumor segmentation. The dataset, which has received several enhancements over the years, consists of preoperative multimodal MRI scans of glioblastoma and lower-grade glioma with ground truth labels and survival data for participants to segment and predict the tumor.	[118]
KiTS	The Kidney Tumor Segmentation Benchmark (KiTS) is a dataset used to evaluate and compare algorithms for kidney tumor segmentation. The dataset consists of CT scans of the kidneys and kidney tumors, with 300 scans in total. The data and segmentations are provided by various clinical sites around the world.	[119]
LiTS	The Liver Tumor Segmentation Benchmark (LiTS) is a dataset used to evaluate and compare liver tumor segmentation algorithms. The dataset consists of CT scans of the liver and liver tumors, with 130 scans in the training set and 70 scans in the test set. The data and segmentations are provided by various clinical sites around the world.	[94]
MURA	The Musculoskeletal Radiographs (MURA) dataset is a large dataset of musculoskeletal radiographs containing 40,561 images from 14,863 studies. Each study is manually labeled by radiologists as either normal or abnormal.	[120]
MedPix	A free online medical image database with over 59,000 indexed and curated images from over 12,000 patients.	[121]
NIH Chest X-rays	A large dataset of chest X-ray images containing over 112,000 images from more than 30,000 unique patients. The images are labeled with 14 common disease labels.	[122]

## Data Availability

Not applicable.

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
