# Peer review of "How Artificial Intelligence Is Shaping Medical Imaging Technology: A Survey of Innovations and Applications"

_bioengineering, 2023, doi:10.3390/bioengineering10121435_

Round 1

Reviewer 1 Report

Comments and Suggestions for Authors

The paper by Pinto-Coelho is very well-written and structured. The topic of this review paper is timely. I will accept the manuscript, but I want to highlight some minor improvements.

1) I will encourage the author to include the study's selection criteria and tools.

2) It will be better to include graphics as well. For example, a figure can be added to show the year-wise technological advancement. More figures related to imaging modalities, types of disease, etc., can be incorporated to make the article more interesting. 

Author Response

Reviewer 1

The paper by Pinto-Coelho is very well-written and structured. The topic of this review paper is timely. I will accept the manuscript, but I want to highlight some minor improvements.

1) I will encourage the author to include the study's selection criteria and tools.

ANS: Thank you for your comments. A section explaining how the review methodology was included as well as a description of the supporting tools.

2) It will be better to include graphics as well. For example, a figure can be added to show the year-wise technological advancement. More figures related to imaging modalities, types of disease, etc., can be incorporated to make the article more interesting.

ANS: Thank you for your comments. Several figures were included for improving content understanding.

Reviewer 2 Report

Comments and Suggestions for Authors

I read the paper. It is an interesting topic, but there are some issues that have to be solved before publication. Following some comments:

·         Introduction section should be reported by literature.

·         Systematic analysis of the review was not performed. Thus, authors should clarify the methos for paper and topic selection.

·         Object detection is an important aspect in the field of medical imaging (e.g., 10.1016/j.engappai.2023.107392, 10.3390/diagnostics13101683, ). I suggest including this topic.

·         Methods should be stratified according to the used images or according to the medical field of interest. I suggest reviewing the format of the entire paper in order to emphasize these aspects.

Author Response

I read the paper. It is an interesting topic, but there are some issues that have to be solved before publication. Following some comments:

  • Introduction section should be reported by literature.

ANS: Thank you for your comment. The introduction was revised and it is now supported by references.

  • Systematic analysis of the review was not performed. Thus, authors should clarify the methos for paper and topic selection.

ANS: Thank you for your comment. The purpose of the article was to provide an overview of the medical imaging landscape that is being shaped by AI. For this purpose, the search methodology was structured as rigorous as possible, though some ad-hoc criteria had to be used since the area is still very recent.

  • Object detection is an important aspect in the field of medical imaging (e.g., 10.1016/j.engappai.2023.107392, 10.3390/diagnostics13101683, ). I suggest including this topic.

ANS: Thank you for your comment. References to the second paper was included as well as a paragraph about the YOLO algorithm. The paragraphs are here transcribed: “Within the broad category of computer vision and artificial intelligence, the YOLO algorithm [42], which stands for "You Only Look Once, has gained a lot of popularity due to its performance in real-time object detection tasks. In the medical imaging field the term "YOLO" is sometimes used more broadly to refer to implementations or systems that use the one of the versions of the YOLO algorithm. It approaches object detection as a regression problem, predicting bounding box coordinates and class probabilities directly from the input image in a single pass through its underlying neural network (composed of backbone, neck and head sections). This single-pass processing, where the image is divided into a grid for simultaneous predictions, distinguishes YOLO from other approaches and contributes to its exceptional speed. Post-prediction, non-maximum suppression is applied to filter redundant and low-confidence predictions, ensuring that each object is detected only once. In the medical field, YOLO has been used for a variety of imaging tasks, including cytology automation [43], detecting lung nodules in CT scans [44], segmentation of structures [45], detecting breast cancer in mammograms [46] or to track needles in ultrasound sequences [47], among others. YOLO’s fast and accurate object detection capabilities make it an excellent choice for many medical imaging applications.”

  • Methods should be stratified according to the used images or according to the medical field of interest. I suggest reviewing the format of the entire paper in order to emphasize these aspects.

ANS: Thank you for your comment. There are several ways to group the articles.  In this paper the focus has been placed on the possibilities opened by AI, as in the title. Focusing on specific medical fields or image types would be very interesting, however we believe that would require a very different approach.

Reviewer 3 Report

Comments and Suggestions for Authors

This manuscript was written as a survey article for, ‘How AI is shaping Medical Imaging Technology’.

1.     The introduction section mentioned here have not cited any articles. Specifically for all the advantages and improvements to the medical technology by Artificial intelligence, listed in the introduction section. It would be good to cite the relevant articles for all the advancements and improvements mentioned in this section.

2.     Section for, ‘Technological innovation’, is more focused on the review and improvement of the Artificial algorithm, this has been mentioned in various articles. Are you focusing in AI innovations in general or for medical technology?

3.     Section for ‘Technological innovation’, Could you elaborate on this line, ‘These, located deep in the network, distill data into compact, meaningful forms that are highly discriminative.’

4.     Section for ‘Technological innovation’, you mentioned that ‘For classification tasks, a softmax activation function is typically used to calculate class probabilities’. Can you cite the relevant articles, and can you elaborate why other activation functions are not used, specifically sigmoid.

5.     Section for ‘Transformer’, this line needs to be elaborated, ‘The class token's state in the ViT's output functions as the representation of the entire image.’

6. Section for ‘Generative networks’, there was no results mention. The results should be added to all the papers / points mentioned in this section, specifically when GANs is compared with other techniques. This will clarify the differences between these approaches.

7. Section for ‘Application’, it would be useful to mention the AI algorithms and approaches that have been used and have shown considerable improvements in the specific (DR, AMD, OCT) fields (retinal diseases).

8. Section for ‘Application’, specifically for breast cancer, radiomic features for the state of art approaches can be listed and explained, also the approaches used can be elaborated for more clarity.

9. Visual aids can be helpful to understand the content of the paper, specifically, results, approaches and architectures used.

10.  Section for ’Application’, specifically for liver cancer, the approach was mentioned, ensembling 4 networks, why was only this approach selected? Is this approach the state of art compared for liver segmentation?

11.  Section for ‘Image and Model Enhancement for Improved Analysis’, Specifically for NDG-CAM, it would be good to mention the results and compare the results with the state of art methodology.

12.  Section for ‘Image and Model Enhancement for Improved Analysis’, specifically for models mentioned here like Vision, Swin transformer, it would be good to mention the results for the listed datasets such as BraTS, LiTS and KiTS.

Comments on the Quality of English Language

English language used was ok and require minor improvement.

Author Response

Reviewer 3

  1. The introduction section mentioned here have not cited any articles. Specifically for all the advantages and improvements to the medical technology by Artificial intelligence, listed in the introduction section. It would be good to cite the relevant articles for all the advancements and improvements mentioned in this section.

ANS: Thank you for your comment. Many references have been added to the introduction section.

  1. Section for, ‘Technological innovation’, is more focused on the review and improvement of the Artificial algorithm, this has been mentioned in various articles. Are you focusing in AI innovations in general or for medical technology?

ANS: Thank you for your comment. The objective was to provide an overview of technological innovations that offer interesting possibilities in the medical image field. For this, an overview of the technological innovations is first provided and then some application cases are covered. This is a very broad objective and it is not possible to provide a deep coverage of all aspects. However we believe that, for the authors, this can be important since many possible research paths are opened.

  1. Section for ‘Technological innovation’, Could you elaborate on this line, ‘These, located deep in the network, distill data into compact, meaningful forms that are highly discriminative.’

ANS: Thank you for your comment. Additional details have been included and are here transcribed: “These, located deep in the network, distill data into compact, meaningful forms that are highly discriminative. Or, in other words, after the progressive extraction of information, layer after layer, raw data is refined into more condensed and abstract representations that are imbued with semantic meaning, encapsulating essential characteristics of the input. They are highly discriminative and have lower dimensionality than the raw input data, which not only conserves computational resources but also simplifies subsequent processing, making it especially beneficial in the analysis of high-dimensional data, such as images.”

  1. Section for ‘Technological innovation’, you mentioned that ‘For classification tasks, a softmax activation function is typically used to calculate class probabilities’. Can you cite the relevant articles, and can you elaborate why other activation functions are not used, specifically sigmoid.

ANS: Thank you for your insightful comment. The text has been revised with additional explanation and with a reference where comparisons are performed. The new text is here transcribed: “For classification tasks, a sigmoid or a softmax activation function is typically used to calculate class probabilities, providing the final output of the CNN [25]. sigmoid is commonly employed in binary classification, producing a single probability score indicating the likelihood of belonging to the positive class. The softmax function is favored for its ability to transform raw output scores into probability distributions across multiple classes. This conversion ensures that the computed probabilities represent the likelihood of the input belonging to each class, with the sum of probabilities equating to 1, thereby constituting a valid probability distribution. Beyond this interpretability, both functions are differentiable, a critical attribute for the application of gradient-based optimization algorithms like backpropagation during training.”

  1. Section for ‘Transformer’, this line needs to be elaborated, ‘The class token's state in the ViT's output functions as the representation of the entire image.’

ANS: Thank you for your comment. More information was added as transcribed: “The class token's state in the ViT's output underscore a pivotal aspect of the model's architecture since it acts as a global aggregator of information from all patches, offering a comprehensive representation of the entire image. The token's state is dynamically updated during processing, reflecting a holistic understanding that encapsulates both local details but also the broader context of the image.”

  1. Section for ‘Generative networks’, there were no results mentioned. The results should be added to all the papers / points mentioned in this section, specifically when GANs is compared with other techniques. This will clarify the differences between these approaches.

ANS: Thank you for your comment. We believe that “results” are focused on advantages and disadvantages of the technologies, more on an overview perspective. Values would be based in a specific context and within the broad scope of this manuscript. An additional paragraph, including a comparison of approaches has been added, and is here transcribed: “GANs are highly popular for magnetic resonance applications due to their ability to generate additional but also due to the existing datasets that can support the training of effective models [74]. Reconstruction and segmentation tasks are also an important field of application. Here, the adversarial training plays a crucial role in imposing robust constraints on both the shape and texture of the generator's output  [74]. In some cases GANs can be preferred over VAE due easier optimal model optimization [75]. In many applications a balance must be found between the ability to generate high-quality samples, achieve fast sampling (inference), and exhibit mode diversity [76].”

  1. Section for ‘Application’, it would be useful to mention the AI algorithms and approaches that have been used and have shown considerable improvements in the specific (DR, AMD, OCT) fields (retinal diseases).

ANS: Thank you for your comment. Additional paragraphs have been included to cover these topics. A full coverage is not possible within the broad scope of the article. The text is here transcribed: “Diabetic retinopathy (DR) is a significant cause of blindness globally, and early detection and intervention can help change the outcomes of the disease.  AI techniques, including deep learning and convolutional neural networks (CNN), have been applied to the analysis of retinal images for DR screening and diagnosis [91]. Some studies have shown promising results in detecting referable diabetic retinopathy (rDR) using AI algorithms, with high sensitivity and specificity compared to human graders [92], while reducing the associated human resources. For example, a study using a deep learning-based AI system achieved 97.05% sensitivity, 93.4% specificity, and 99.1% area under the curve (AUC) in classifying rDR as moderate or worse diabetic retinopathy, referable diabetic macular edema, or both [92]. Nevertheless, there are al-so shortcomings, such as the lack of standards for development and evaluation and the limited scope of application [93]. AI can also help in the detection and prediction of age-related macular degeneration (AMD). AI-based systems can screen for AMD and predict which patients are likely to progress to late-stage AMD within two years [94]. AI algorithms can provide analyses to assist physicians in diagnosing conditions based on specific features extrapolated from retinal images [95]. Yet in this area, Optical Coherence Tomography (OCT) is a valuable tool in diagnosing various eye conditions, and where artificial intelligence (AI) can successfully be used. AI-assisted OCT has several advantages and applications in ophthalmology for diagnosis, monitoring and disease progression estimation (for e.g. glaucoma, macular edema, or age-related macular degeneration) [96]. AI-assisted OCT can provide more accurate and sensitive results compared to traditional methods [97]. For example, an OCT-AI-based telemedicine platform achieved a sensitivity of 96.6% and specificity of 98.8% for detecting urgent cases, and a sensitivity of 98.5% and specificity of 96.2% for detecting both urgent and routine cases [98]. These tools can lead to more efficient and objective ways of diagnosing and man-aging eye conditions.”

  1. Section for ‘Application’, specifically for breast cancer, radiomic features for the state of art approaches can be listed and explained, also the approaches used can be elaborated for more clarity.

ANS: Thank you for your insightful comment. The important area of radiomics has been asl oincluded. The text is here transcribed: “Radiomics and artificial intelligence (AI) play pivotal roles in advancing breast cancer imaging, offering a range of applications across the diagnostic spectrum. These technologies contribute significantly to risk stratification, aiding in the determination of cancer recurrence risks and providing valuable insights to guide treatment decisions [89,90]. Moreover, AI algorithms leverage radiomics features extracted from diverse medical imaging modalities such as mammography, ultrasound, magnetic resonance imaging (MRI), and positron emission tomography (PET) to enhance the accuracy of detecting and classifying breast lesions [89,90]. For treatment planning, radiomics furnishes critical information regarding treatment effectiveness, facilitating the prediction of treatment responses and the formulation of personalized treatment plans [91]. Additionally, radiomics serves as a powerful tool for prognosis, enabling the prediction of outcomes such as disease-free survival and recurrence risk in breast cancer patients [92]. Furthermore, It has been highlighted the robustness of MRI-based radiomics features against inter-observer segmentation variability, indicating their potential for future breast MRI-based radiomics research [93].”

  1. Visual aids can be helpful to understand the content of the paper, specifically, results, approaches and architectures used.

ANS: Thank you for your important comment. Several diagrams and charts have been included to illustrate some of the concepts and to better describe some data.

  1. Section for ’Application’, specifically for liver cancer, the approach was mentioned, ensembling 4 networks, why was only this approach selected? Is this approach the state of art compared for liver segmentation?

ANS: Thank you for your comment. The paper was selected due to its original application of several AI technologies with interesting results. Additional text was included with references to review papers where the reader can gain further knowledge in this specific topic. The text is here transcribed: “Liver cancer is the third most common cause of death from cancer worldwide [89] and its incidence has been growing. Again, the development of the disease is often asymptomatic, making screening and early detection crucial for a good prognosis. In [8], the authors focus on the segmentation of liver lesion in CT images of the LiTS dataset [90]. As a novelty, the paper proposes an intelligent decision system for segmenting liver and hepatic tumors by integrating four efficient neural networks (ResNet152, ResNeXt101, DenseNet201, and InceptionV3). These classifiers are independently operated and a final result is obtained by postprocess to eliminate artifacts. The obtained results were better than those obtained by the individual networks. In fact, concerning liver and pancreatic images the use of AI algorithms is already a reality for speeding up repetitive tasks such as segmentation, acquire new quantitative parameters such as lesion volume and tumor burden, improve image quality, reduce scanning time, and optimize imaging acquisition [91]”.

  1. Section for ‘Image and Model Enhancement for Improved Analysis’, Specifically for NDG-CAM, it would be good to mention the results and compare the results with the state of art methodology.

ANS: Thank you for your comment. NDG-CAM also has additional information, as transcribed: “In complex healthcare scenarios, it is crucial for clinicians and practitioners to understand the reasoning behind AI models' predictions and recommendations. Explainable AI (XAI) plays a pivotal role in the domain of medical imaging techniques for decision support, where transparency and interpretability are paramount. In [9], the authors address the problem of nuclei detection in histopathology images, which is a crucial task in digital pathology for diagnosing and studying diseases. They specifically propose a technique called NDG-CAM (Nuclei Detection in Histopathology Images with Semantic Segmentation Networks and Grad-CAM). Grad-CAM (Gradient-weighted Class Activation Mapping) is a technique used in computer vision and deep learning to visualize and interpret the regions of an image that are most influential in the prediction made by a convolutional neural network. Hence, in the proposed methodology, the semantic segmentation network aims to accurately segment the nuclei regions in histopathology images, while Grad-CAM helps visualize the important regions that contribute to the model's predictions, helping to improve the accuracy and interpretability of nuclei detection. The authors compare the performance of their method with other existing nuclei detection methods, demonstrating that NDG-CAM achieves improved accuracy while providing interpretable results.”

  1. Section for ‘Image and Model Enhancement for Improved Analysis’, specifically for models mentioned here like Vision, Swin transformer, it would be good to mention the results for the listed datasets such as BraTS, LiTS and KiTS.

ANS: Thank you for your comment. A new subsection covering some relevant datasets was included in the applications section. Fully covering this topic would demand another paper about “Machine Learning datasets for medical imaging”.

Round 2

Reviewer 2 Report

Comments and Suggestions for Authors

The authors solved my issues.

Comments on the Quality of English Language

I suggest slightly revising the english